Network analysis of depression and anxiety symptoms in Chinese rheumatoid arthritis patients

Zhang Lijuan 1 2
Zhu Weiyi venyzwy@163.com 1
Wu Beiwen Gaoan2005new@163.com 1
1 Department of Nursing, Ruijin Hospital, Shanghai Jiao Tong University School of Medicine , Shanghai , China
2 School of Nursing, Shanghai Jiao Tong University School of Medicine , Shanghai , China
Khosravi Mohsen
Electronic publication date: 2023 Nov 6
Publication date: 2023
Volume: 11
Electronic Location ID: e16356
Received 2023 Jul 14; Accepted 2023 Oct 4
Copyright: ©2023 Zhang et al.
Copyright year: 2023
Copyright holder: Zhang et al.
License: This is an open access article distributed under the terms of the Creative Commons Attribution License, which permits unrestricted use, distribution, reproduction and adaptation in any medium and for any purpose provided that it is properly attributed. For attribution, the original author(s), title, publication source (PeerJ) and either DOI or URL of the article must be cited.
License URL: https://creativecommons.org/licenses/by/4.0/

Keywords: Rheumatoid arthritis, Anxiety, Depression, Quality of life, Network analysis

Funding: The Chinese National Natural Science Foundation 71904118 Innovative Research Team of High-level Local Universities in Shanghai SHSMU-ZDCX20210602 This study was supported by grants from the Chinese National Natural Science Foundation (Grant No.71904118) and the Innovative Research Team of High-level Local Universities in Shanghai (Grant No. SHSMU-ZDCX20210602). The funders had no role in study design, data collection and analysis, decision to publish, or preparation of the manuscript.

==============================
Background

Rheumatoid arthritis (RA) patients are susceptible to comorbid anxiety and depression. From the network model perspective, comorbidity is due to direct interactions between depression and anxiety symptoms. The objective of this study was to assess the network structure of depression and anxiety symptoms in Chinese RA patients and identify the central and bridge symptoms as well as how depression and anxiety symptoms are related to quality of life (QoL) in the network.

Methods

A total of 402 Chinese RA patients were included in this study. Depression and anxiety symptoms were measured by the Hospital Anxiety and Depression Scale (HADS). R software was used to estimate the network. Specifically, we computed the predictability, expected influence (EI) and bridge expected influence (BEI) for each symptom and showed a flow network of “QoL”.

Results

Our network revealed that the strongest edge was D2 “See the bad side of things” and D3 “Not feeling cheerful” across the whole network. For centrality indices, D3 “Not feeling cheerful” and D6 “Feeling down” had the highest EI values in the network, while A4 “Trouble relaxing” and D6 “Feeling down” had the highest BEI values of their respective community. As to “QoL”, the strongest direct edge related to it was A1 “Nervousness”.

Conclusions

“Feeling down” and “Not feeling cheerful” emerged as the strongest central symptoms, while “Trouble relaxing” and “Feeling down” were bridge symptoms in the anxiety-depression network of RA patients. Intervention on depression and anxiety symptoms in nurses should prioritize these symptoms.

Introduction

Rheumatoid arthritis (RA) is a chronic autoimmune inflammatory disease that affects 1% of the global population (Potempa, Mydel & Koziel, 2017). Chronic painful state of disease and undefined period of treatment severely imperil psychological well-being (Lwin et al., 2020), among which depression and anxiety are the most common mental disorders. A previous studies (Itaya et al., 2021; Juárez-Rojop et al., 2020; Katchamart et al., 2020) have demonstrated that 26–46% of RA patients have anxiety symptoms, and 14.8–34.2% have depression symptoms. A systematic review and meta-analysis (Ng et al., 2022) including 11 cohort studies found that patients with depression also had a 34% greater risk of developing RA, which indicated that there existed a bidirectional association between RA and depression. Growing evidence demonstrated the negative effects of anxiety and depression in the context of RA, including the low RA remission rate and treatment response (Boer, Huizinga & van der Helm-van Mil, 2019), the increasing health care costs, disability (Ji et al., 2017) and mortality (Pinho de Oliveira Ribeiro et al., 2013), and poor quality of life (QoL) (Khan et al., 2021; Zhang et al., 2017). RA patients had a 47% greater risk of incident depression compared to controls, These data highlight the importance of managing anxiety and depression in RA patients, not only to improve psychiatric health, but also to aid in management of physical aspects of the disease.

It has been revealed that depression and anxiety are highly comorbid (Lamers et al., 2011; Winer et al., 2017). According to the results from an epidemiological survey (Andrade et al., 2003) conducted in ten countries, one third to one-half of participants with depression also reported having a history of anxiety. It is known that anxiety and depression tend to trigger each other (Jacobson & Newman, 2017). In other words, when one disorder is present, the onset of another disorder may rise correspondingly (Wittchen et al., 2000). The level of comorbidities is closely associated with a range of negative consequences (Belzer & Schneier, 2004; Olfson et al., 1997), including the severity of the illness, chronicity, and poor prognosis. Taken together, it is necessary to identify underlying mechanisms and develop targeted intervention strategies of these two co-occurring conditions because simply targeting only one disorder may not be effective.

The most popular method of previous studies focusing on anxiety, depression, and their comorbidity often based on sum-scores, failing to point out the difference and the interaction within the symptoms (Katchamart et al., 2020; Peterson et al., 2019; Zhang et al., 2017). In recent years, the network analysis has been used as an important way of analyzing individual symptoms and their interactions, which provides a promising tool for understanding the symptomology of psychiatric disorders. It is a data-driven approach to analyze the relationship between complex variables from a mathematical point of view and display it intuitively (Beard et al., 2016; Mangion et al., 2022). The network is composed of two parts, one is the node, which represents the symptoms, and the other is the edge, which represents the relationship between the symptoms (Robinaugh, Millner & McNally, 2016). Compared with the traditionally correlational analysis, network model can provide the corresponding centrality and predictability index for each node to examine its importance and controllability in the whole network (Contreras et al., 2019; Haslbeck & Waldorp, 2018). Furthermore, it evaluates interconnectedness among different disorders in the nerwork by exploring the bridge symptoms (Jones, Ma & McNally, 2021). The network model considers that activation of these bridge symptoms is likely to result in development and maintenance of both disorders. Core and bridge symptoms represent promising and effective targets for intervention and treatment (Robinaugh, Millner & McNally, 2016).

An increasing number of studies have employed the network analysis to assess anxiety and depression among different populations, including HIV patients (Liu et al., 2022), functionally impaired elderly (Yang et al., 2022), migrant Filipino domestic workers (Garabiles et al., 2019), Chronic Obstructive Pulmonary Disease (COPD) (Yohannes et al., 2022) and nursing students during the COVID-19 pandemic (Peng et al., 2022), which documented both common and unique depression-anxiety network across different populations. Despite the high interest in the prevalence and correlation of mental disorders among RA patients (Juárez-Rojop et al., 2020; Katchamart et al., 2020), no study has assessed them at a symptom level applying network model. Due to the different study populations and measurement tools, and the data-driven nature of the network approach, the above findings often have limitations and are difficult to generalize to patients with RA. To our knowledge, no research has explored the depression and anxiety using the network analysis in Asian RA patients. It has been reported that culture may shape the presentation of depression and anxiety symptoms (Chentsova-Dutton & Tsai, 2009). For instance, Asian individuals tend to believe that negative emotions have cognitive and motivational utility and are less likely to engage in hedonic emotion regulation. Therefore, we conducted the current study to map out the network structure of anxiety-depression symptoms in Chinese RA patients. Herein, we want to identify the central and bridge symptoms within the depression-anxiety network and foster greater clarity on how best to treat depression and anxiety. In addition, QoL is considered as one of the main outcome measures for RA patients. A previous study (Zhang et al., 2017) has demonstrated that depression and anxiety were significantly associated with all dimensions of SF-36 in Chinese RA patients. Therefore, we particularly focus on the symptoms that are directly related to “QoL”, in order to provide reliable evidence for improving RA patients’ QoL.

Materials & Methods

Participants and procedure

When patients came to the hospital for RA-related issues, they were asked if they want to participate in the study. The inclusion criteria were adults, aged 18 years and older with a diagnosis of RA fulfilling either the American College of Rheumatology (ACR) 1987 revised criteria for the classification of RA (Arnett et al., 1988) or the 2010 RA classification criteria (Aletaha et al., 2010). A total of 402 patients were recruited from the Department of Rheumatology and Immunology, Shanghai Ruijin Hospital from January 2022 to May 2023. The exclusion criteria were the following: (1) they did not fulfill the ACR 1987 revised criteria or the 2010 RA classification criteria; (2) age 18 years old; (3) they refused to participate in this study; (4) they had co-morbidities such as serious cardiac, respiratory, or endocrine disease that might influence mental health. This study was approved by the Ethics Committee of the Ruijin Hospital, Shanghai Jiao Tong University School of Medicine, and all participants provided written informed consent.

Measurements

Demographic and disease characteristics

When patients came to the hospital for RA-related issues, clinical interview, clinical examination and questionnaires were collected from them at the same time. Demographic and disease characteristics were collected as previously described in Ji et al. (2017). Age, gender, body mass index (BMI), marital status, education, employment, income/family/year, health insurance, comorbid condition, disease duration, family history, hospitalization history, tobacco and alcohol usage, the use of disease modifying anti-rheumatic drugs (DMARDs), nonsteroidal anti-inflammatory drugs (NSAIDs), corticosteroids, and biologics were all recorded. The 28-joint Disease Activity Score (DAS28) was used to assess disease activity by two rheumatologists. Differences were settled by consensus. DAS28 is a continuous measure of RA disease activity that combines information from 28 swollen joint counts, 28 tender joint counts, the rate of erythrocyte sedimentation rate (ESR), as well as the patient’s recognition of disease activity from 0 cm (not active at all) to 10 cm (very active) (Prevoo et al., 1995). Furthermore, the 10-cm horizontal visual analogue score (VAS) (0 = no pain, and 10 = most severe pain) was used to assess pain (Hawker et al., 2011). The Health Assessment Questionnaire (HAQ) was used to assess functional capacity, and lower scores (ranging 0–3) indicated high functional capacity (Fries, 1991).

Depression and anxiety

The depression and anxiety were assessed using the Hospital Anxiety and Depression Scale (HADS) (Covic et al., 2012). It was a 14-item questionnaire and each item had a 4-point Likert scale and was scored between 0 and 3. Anxiety and depression were scored separately using the 7-item subscales, and each items with higher scores indicated severer anxiety or depression symptoms. The Chinese version of HADS had acceptable internal consistency and test-retest reliability, with a Cronbach alpha of 0.85 and intraclass correlation coefficient of 0.90, respectively (Yang et al., 2014).

Quality of life

Participants’ health status was assessed using the Short Form 36 health survey (SF-36) in the past 4 weeks (Brazier et al., 1992). The total scores range from 0 to 100, and higher scores indicates better health status. Its eight subscales include: physical functioning (PF, 10 items), role-physical functioning (RP, four items), bodily pain (BP, two items), general health perception (GH, five items), vitality (VT, four items), social functioning (SF, two items), role-emotional functioning (RE, three items), and mental health (MH, five items). It was summarized in two summary domains: the physical (PCS) and mental component summary (MCS). The total score of QoL used in this study is the summary of PCS and MCS. The Chinese version of SF-36 has a Cronbach’s alpha of 0.85 to 0.87, showing good internal consistency reliability; The intraclass correlational coefficients were above 0.7; the split-half reliability coefficient of the SF-36 is 0.814, indicating good reliability (Wu et al., 2023).

Statistical analysis

All the statistical analysis was done with RStudio (RStudio Team, 2022). The baseline characteristics are described using frequencies (percentages), and means (standard deviations).

Network analysis

We applied R package qgraph to estimate the Gaussian graphical model (GGM) (Epskamp et al., 2018b) of the comorbidity network in Chinese RA patients. In order to obtain a more stable and easier to interpret sparse network, the least absolute shrinkage and selection operator (LASSO) regularization and the extended Bayesian information criterion (EBIC) was used to shrink all edges and set the edge with small partial correlation exactly to zero (Epskamp & Fried, 2018; Friedman, Hastie & Tibshirani, 2008). Moreover, we set the tuning parameter to 0.5, which get a good balance between the sensitivity and specificity of extracting true edge (Epskamp & Fried, 2018). Considering the ordinal nature of the HADS, the Spearman rho correlation method was employed in the network construction (Epskamp & Fried, 2018). The nodes in the network represented items of HADS and were grouped into the anxiety and depression community according to their theoretical sources (Yang et al., 2014). The relationships of symptoms were presented by edges, and thicker edges indicated higher correlations (Robinaugh, Millner & McNally, 2016). The visualization of the network was conducted by the Fruchterman-Reingold algorithm.

We estimated the centrality indices “expected influence” (EI) of nodes to identify which symptoms are more important in the network (Burger et al., 2023). In addition, we used R-package networktools to calculate the bridge expected influence (BEI) for each node and then identified the bridge symptoms (Jones, Ma & McNally, 2021). Bridge symptoms were chosen with an 90th percentile BEI threshold. Furthermore, Rpackage mgm was used to compute the predictability for each node (Haslbeck & Waldorp, 2018). Node with high predictability means that it may be easily influenced by its neighbors. The graphics function “flow” of the R package qgraph was applied to intuitively display the anxiety and depression symptoms directly or indirectly related to the “QoL”. To further assess the robustness of network, we employed non-parametric bootstrapping with 1,000 bootstrap samples to evaluate the accuracy of edge weights by computing 95% confidence intervals through the “bootnet” packages (Epskamp, Borsboom & Fried, 2018a). The case-dropping bootstrap approach was used to test the stability of EI and BEI. The stability of the network was examined by the correlation stability coefficient (CS-C). The value of CS-C preferably should be above 0.5 and should not be below 0.25 (Armour et al., 2017).

Results

Descriptive statistics

Four hundred and twenty-two RA patients took part in this study. However, seven patients did not fulfill the ACR 1987 revised criteria or the 2010 RA classification criteria and thirteen patients declined to participate in this study. Therefore, 402 patients were finally included in this study. Table 1 presents the demographic characteristics of the participants. The mean age of the participants was 55.74 ± 12.76 years old, ranging from 18 to 82 years old. The majority of participants were female (330, 82.1%), married (356, 88.5%), and 68.7% of these patients (276) were unemployed. The mean disease duration and HAQ score was 7.43 ± 8.66 years and 0.16 ± 0.84, respectively. The means (standard deviations), EIs, BEIs and predictability of the symptoms in the anxiety-depression network of RA patients are shown in Table 2.

Table 1 Characteristics of RA patients (n = 402).

Characteristics	n (%), Mean ± SD	
Age, years	55.74 ± 12.76	
Gender, Female	330 (82.1%)	
BMI, kg/m2	22.53 ± 3.43	
Marital status		
Single	22 (5.5%)	
Married	356 (88.5%)	
Others	25 (6.0%)	
Education		
Primary and below	208 (51.7%)	
Secondary	135 (33.6%)	
Graduate and above	59 (14.7%)	
Employment		
Unemployed	276 (68.7%)	
Employed	126 (31.3%)	
Income/family/year		
≤13000 yuan	143 (35.6%)	
13000–33000 yuan	95 (23.6%)	
≥33000 yuan	164 (40.8%)	
Health insurance, yes	372 (92.5%)	
Comorbid condition, yes	226 (56.2%)	
Disease Duration, years	7.43 ± 8.66	
Family history, yes	49 (12.2%)	
Hospitalization history, yes	232 (57.7%)	
Tobacco usage, yes	33 (8.2%)	
Alcohol usage, yes	55 (13.7%)	
DAMARDs usage, yes	313 (77.9)	
NSAIDs usage, yes	66 (16.4)	
Corticosteroids usage, yes	100 (24.8)	
Biologics usage, yes	101 (25.1)	
Pain score	4.33 ± 2.74	
DAS28 (ESR)	3.53 ± 1.42	
HAQ score	0.16 ± 0.84	
Notes.

RA rheumatoid arthritis

SD Standard deviation

BMI Body mass index

DAMARDs disease modifying anti-rheumatic drugs

NSAIDs nonsteroidal anti-inflammatory drugs

DAS28 Disease Activity Score in 28 joints

ESR erythrocyte sedimentation rate

HAQ health assessment questionnaire

Table 2 Descriptive statistics of the HADS items (n = 402).

Items content	Abbreviation	Mean ± SD	EI	BEI	Pre	
Anxiety symptoms						
A1: Feeling tense or ‘wound up’.	Nervousness	0.81 ± 0.72	0.79	0.29	36.1%	
A2: Having frightened feeling as if something awful is about to happen.	Afraid something will happen	0.83 ± 0.96	0.99	0.11	44.6%	
A3: Worrying thoughts go through mind.	Excessive worry	0.66 ± 0.87	0.91	0.51	48.5%	
A4: Not sitting at ease and feeling relaxed.	Trouble relaxing	0.81 ± 0.79	0.68	0.59	31.3%	
A5: Feeling restless.	Restlessness	0.87 ± 0.97	0.43	0.02	17.2%	
A6: Getting sudden feelings of panic.	Sudden panic	0.63 ± 0.81	0.96	0.30	39.6%	
A7: Having frightened feeling like ‘butterflies’ in the stomach.	Afraid of organ damage	0.59 ± 0.71	0.65	0.29	33.5%	
Depression symptoms						
D1: Not enjoying the things that used to enjoy.	Not feeling interested	0.98 ± 1.04	0.63	0.12	28.9%	
D2: Unable to laugh and see the funny side of things.	See the bad side of things	0.72 ± 0.91	1.00	0.20	44.7%	
D3: Not feeling cheerful.	Not feeling cheerful	0.62 ± 0.82	1.10	0.51	52.9%	
D4: Lost interest in appearance.	Lost interest in appearance	1.21 ± 1.15	0.25	0.03	10.1%	
D5: Not looking forward with enjoyment to things.	Feeling pessimistic	0.57 ± 0.82	0.98	0.15	47.0%	
D6: Feeling being slowed down.	Feeling down	0.71 ± 0.69	1.10	0.80	51.4%	
D7: Not enjoying good books or radio or TV program.	Not enjoying books or programs	0.50 ± 0.80	0.52	0.30	18.2%	
Notes.

HADS Hospital Anxiety and Depression Scale

SD Standard deviation

EI expected influence

BEI bridge expected influence

Pre Predictability

Structure of the anxiety-depression network

Figure 1A displayed the network structure of anxiety and depression in Chinese RA patients. Theoretically, there are up to 91 possible edges in the network, while our result showed only 60 non-zero edges. All edges were positive, among which there were 49 edges (53.85%) across communities and 42 edges (46.15%) within communities. The strongest edge within the anxiety symptoms was A2 “Afraid something will happen” and A6 “Sudden panic” (edge weight = 0.53), followed by A1 “Nervousness” and A2 “Afraid something will happen” (edge weight = 0.51). The strongest edge within the depression symptoms were D2 “See the bad side of things” and D3 “Not feeling cheerful” (edge weight = 0.60), and D2 “See the bad side of things” and D5 “Feeling pessimistic” (edge weight = 0.59). A3 “Excessive worry” and D3 “Not feeling cheerful” (edge weight = 0.54) showed the strongest association between anxiety and depression symptoms (Fig. S2). More detailed information of the network structure was summarized in Fig. S1. Moreover, predictability is presented as a circle around a node in Fig. 1A. Table 2 showed that the node predictability values ranged from 10.1% to 52.9%. D3 “Not feeling cheerful”, and A3 “Excessive worry” had the modest predictability, showing that 52.9% and 48.5% of their variance can be explained by their neighbors.

Figure 1 (A–B) The anxiety-depression network structure and the expected influence indices in Chinese RA patients.

Figure 2 (A-B) The anxiety-depression network structure highlighting the bridge symptoms and the bridge expected influence indices in Chinese RA patients.

The central and bridge symptoms

According to Fig. 1B, D3 “Not feeling cheerful” and D6 “Feeling down” were identified as core symptoms, because they had the highest EI values (1.10, 1.10). The centrality differs test indicated that the two nodes (D3 and D6) were statistically stronger than other nodes in the network (Fig. S3). Figure 2A showed the centrality of bridge symptoms of the two clusters. As displayed in Fig. 2B, A4 “Trouble relaxing” (0.59) and D6 “Feeling down” (0.80) had the highest BEIs in their own community, as a result, they were recognized as bridge symptoms that drove the comorbid depression and anxiety symptoms. In addition, the BEIs of A4 “Trouble relaxing” and D6 “Feeling down” were significantly larger than the other symptoms in the network (Fig. S4). Figure 3 showed that the CS coefficient of EI was 0.672, representing ideal stability of EI index. The CS coefficient of BEI was 0.672, and it represented ideal stability of BEI indices. The result of the bootstrapped 95% CIs were narrow, which indicated high accuracy (Fig. 4).

Figure 3 The stability of the depression-anxiety network.

The stability of central and bridge expected influence by case-dropping bootstrap.

Figure 4 The accuracy of the network edges by non-parametric bootstrapping.

The gray area represents the bootstrap 95% confidence interval.

Flow network of QoL

Figure 5 depicted a flow diagram showing how QoL “quality of life” was related to all other symptom of the network. It is obvious that seven symptoms were directly related to QoL “quality of life”, while seven symptoms indirectly related to it. Among the direct edges with QoL “quality of life”, the strongest edges were with A1 “Nervousness” (edge weight = −0.533), and D1 “Not feeling interested” (edge weight = −0.443).

Figure 5 The flow network of quality of life in Chinese RA patients.

Discussion

This is the first study to explore the network of anxiety and depression symptoms among Chinese RA patients. The central symptoms of the network were “Not feeling cheerful” and “Feeling down”. “Trouble relaxing” and “Feeling down” served as bridge symptoms linking anxiety and depression. Additionally, nervousness was significantly associated with QoL.

The complex relationships between anxiety and depression

The nodes and edges in the network structure displayed the complex relationships among anxiety and depression symptoms, and these relationships may result in comorbidity and show the potential interactions of anxiety-depression in Chinese RA patients. Thus, the important edges linking the anxiety and depression communities should be discussed. In the present study, A1 “Nervousness” was positively correlated with A2 “Afraid something will happen” and D6 “Feeling down”. To our knowledge, this result has not been discovered in previous studies. This finding may be unique to the specific sample in present study, which needs to be further investigated. It could be explained that the nervousness of chronic disease and the fear of the undefined period of treatment may severely result in “Feeling down” in Chinese RA patients. We also found a strong relationship among D2 “See the bad side of things”, D3 “Not feeling cheerful” and D5 “Feeling pessimistic”, which has been confirmed by previous study (Liu et al., 2022). Furthermore, D6 “Feeling down” and positively correlated with A3 “Excessive worry” and A6 “Sudden panic”. Previous studies (Hedman-Lagerlöf, Hedman-Lagerlöf & Öst, 2018; Lim & Tierney, 2023) have suggested that mindfulness-based interventions and positive thought-based therapies may be effective in enhancing the levels of perceived cheerfulness and reducing panic related symptoms and the inability to positive. The Adaptive Information Processing model and the therapeutic hierarchy have been shown to reduce worry related symptoms (Gierus, 2020).

The central and bridge symptoms of anxiety and depression

The higher value of the EI, its association with other symptoms in the network is stronger (Burger et al., 2023). Consequently, EI may play an important part in identifying symptoms that activate or maintain psychopathological networks as well as providing potential intervention targets. In the current study, we found that D3 “Not feeling cheerful” and D6 “Feeling down” were the core symptoms in the anxiety-depression network. This centrality result is in line with previous study exploring symptom networks for depression and anxiety of the HIV-positive people (Liu et al., 2022). Bridge expected influence centrality is essential for calculating the co-occurrence of mental disorders. It helps to understand the links between comorbidities and provides potential targets for intervention (Jones, Ma & McNally, 2021). Interestingly, we also demonstrated that “Feeling down” was the most critical bridge symptom in the depression-anxiety clusters. It is noteworthy because many RA patients may comorbid depression and anxiety due to joint pain (Fakra & Marotte, 2021), fatigue (Nikolaus et al., 2013), intolerance of medications (Fakra & Marotte, 2021), and sleep problems (Hughes et al., 2021). Therefore, interventions targeting “Feeling down” might generally reduce both anxiety and depression symptoms in Chinese RA patients. In addition, “Trouble relaxing” was another critical bridge symptom in the network. Similar results have been reported in different populations (Liu et al., 2022). According to previous studies (Astin et al., 2002; DiRenzo et al., 2018; Hedman-Lagerlöf, Hedman-Lagerlöf & Öst, 2018; Knittle, Maes & de Gucht, 2010; Nagy et al., 2023), positive psychology interventions such as mindfulness interventions, optimistic interventions, strength-building measures, and cognitive behavior interventions are beneficial to target “Not feeling cheerful”, “Feeling down”, and “Trouble relaxing”. Therefore, positive psychology could be applied to improve RA patients’ cheerfulness, relaxation, mental health, and positive emotions.

QoL improvement for RA patients

As shown in our flow network, “OoL” was directly connected with half symptoms of depression and anxiety. QoL is considered as one of the main outcome measures for RA patients and mental health represents a central component of QoL in RA patients (Matcham et al., 2014). Previous studies (Matcham et al., 2014; Zhang et al., 2017) have confirmed that anxiety and depression were associated with QoL in RA patients. However, there is no study using network analysis evaluated the relationships of anxious and depressive symptoms with QoL in Chinese RA patients; this study might provide initial insights into this problem. The strongest direct edge of “QoL” was with A1 “Nervousness” which has not been reported in previous studies. Through observing and alleviating the “Nervousness”, it might be efficient to improve QoL in Chinese RA population.

Limitations

There were several limitations in the present study. First, we recruited Chinese RA patients who reporting symptoms of depression and anxiety that span the full range of normal to abnormal, which likely limits the generalizability of our findings. Second, the cross-sectional study design didn’t permit causal inferences. Our study can only provide potential hypothesis for future longitudinal and intervention study. Third, the single-item, self-assessment tools rather than a clinical diagnostic interview were used in our study, which might impact the accuracy of our results. Fourth, the network structure in this study was specific to the questionnaires we used. There exist some differences among self-report scales for assessing depression and anxiety, which might impact the repeatability of our results. Fifth, we did not include some potential influential factors such as clinical and psychosocial factors (Ho et al., 2011), biological factors (Liu, Ho & Mak, 2012), or illness perception (Lu et al., 2022) in the network analyses, despite its potential confounder effect. Sixth, we did not describe data about rheumatoid factor (RF), anticitrullinated protein antibodies (anti-CCP), ESR or C-reactive protein (CRP). Finally, depression and anxiety symptoms were significantly related to substance usage or other psychiatric illness which were not assessed in the current study. Network analysis including these factors may be the direction of future research, which could provide new insights into the prevention and intervention of mental health in this population.

Conclusions

In summary, our study developed the network structure of anxiety and depression symptoms among Chinese RA patients. In this network, “Feeling down” was identified as both the strongest central symptom and bridge symptom. “Not feeling cheerful” and “Trouble relaxing” are another core symptoms and bridge symptom in the network, respectively. Furthermore, “Nervousness” was identified as key priority due to its significant association with QoL. These key symptoms, especially “Feeling down” symptom is of great significance in the prevention and treatment of anxiety and depression in RA patients. The flow network shows the association among anxiety and depression symptoms and QoL and provides important suggestions for improving QoL in RA patients.

Supplemental Information

Supplemental Information 1 Raw data of this study

Click here for additional data file.

Supplemental Information 2 Correlation matrix of HADS items

Click here for additional data file.

Supplemental Information 3 Estimation of edge difference within the depression-anxiety network by bootstrapped difference test

Click here for additional data file.

Supplemental Information 4 Estimation of node expected difference within the depression-anxiety network by bootstrapped difference test

Click here for additional data file.

Supplemental Information 5 Estimation of bridge expected difference within the depression-anxiety network by bootstrapped difference test

Click here for additional data file.

The authors appreciate all participants for their cooperation in this study.

Additional Information and Declarations

Competing Interests

Author Contributions

Human Ethics

Data Availability

The authors declare there are no competing interests.

Lijuan Zhang performed the experiments, analyzed the data, prepared figures and/or tables, and approved the final draft.

Weiyi Zhu analyzed the data, prepared figures and/or tables, and approved the final draft.

Beiwen Wu conceived and designed the experiments, authored or reviewed drafts of the article, and approved the final draft.

The following information was supplied relating to ethical approvals (i.e., approving body and any reference numbers):

the Ethics Committee of the Ruijin Hospital, Shanghai Jiao Tong University School of Medicine.

The following information was supplied regarding data availability:

The raw measurements are available in the Supplementary File.

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
