# Peer review of "Network analysis of depression and anxiety symptoms in Chinese rheumatoid arthritis patients"

_PeerJ, doi:10.7717/peerj.16356_

## Round 0.1 · original submission · Major Revisions

I have now received the reviewers' comments on your manuscript. They have suggested some revisions to your manuscript. Therefore, I invite you to respond to the reviewers' comments and revise your manuscript.

Reviewer 1 has suggested that you cite specific references. You are welcome to add it/them if you believe they are relevant. However, you are not required to include these citations, and if you do not include them, this will not influence my decision.

Reviewer 1 ·

Basic reporting

The English is good.

Experimental design

The research methodology is sound.

Validity of the findings

Under Introduction, please discuss the finding of this landmark study:

Elucidating a bidirectional association between rheumatoid arthritis and depression: A systematic review and meta-analysis. J Affect Disord. 2022 Aug 15;311:407-415. doi: 10.1016/j.jad.2022.05.108. Epub 2022 May 25. PMID: 35642835.


The model lacks the following factors and this is a major limitation and needs to address and discuss under discussion:

1) Role of Psychosocial factors in depression and RA:
Clinical and psychosocial factors associated with depression and anxiety in Singaporean patients with rheumatoid arthritis. Int J Rheum Dis. 2011 Feb;14(1):37-47. doi: 10.1111/j.1756-185X.2010.01591.x. Epub 2011 Jan 24. PMID: 21303480.

2) Role of biological factors - IL-17 in depression and RA:
The role of interleukin (IL)-17 in anxiety and depression of patients with rheumatoid arthritis. Int J Rheum Dis. 2012 Apr;15(2):183-7. doi: 10.1111/j.1756-185X.2011.01673.x. Epub 2011 Sep 21. PMID: 22462422.

3) The role of illness perception:
Effects of illness perception on negative emotions and fatigue in chronic rheumatic diseases: Rumination as a possible mediator. World J Clin Cases. 2022 Dec 6;10(34):12515-12531. doi: 10.12998/wjcc.v10.i34.12515. PMID: 36579115; PMCID: PMC9791537.

·

Basic reporting

1. Line 102: “BMI” – this abbreviation is not explained. Please check abbreviation management and correct were necessary.
2. Line 45: “including the low RA remission and treatment response” – “rate” is missing, please add it: “the low RA remission rate and treatment response”;
3. Line 75: “including patients with HIV-positive people” – the words are redundant, please change to something like this: “including HIV patients”;
Line 96-97: “…influence mental disorders; This cross-sectional study…” – the use of a semicolon is wrong, please change it to a full stop.
4. Line 91: “Patients who fulfilled the American College of Rheumatology (ACR) criteria for RA (de Launay et al. 2012)” – please note that de Luanay et al. did not author the ACR criteria or any classification criteria. See also line 105, where you mention the DAS28 which is not published by van Riel & Renskers in 2016. This means you probably have a major problem with your references. Please check and correct all the references and their order in the text.
5. Line 91 “American College of Rheumatology (ACR) criteria for RA” – this is surprising or wrongly reported. ACR issued classification criteria in 1987. Did you use these 1987 criteria? More probably, the 2010 classification criteria were used, but they were issued with EULAR. Please check and update text accordingly.

Experimental design

1. Line 91: “Patients” – what patients are these? All of the patients in the hospital database? Some of them? If they are a sample of the hospital’s RA patient pool, how were they selected to be invited to the study? This information should be briefly stated in the text, since it pertains to inclusion randomness and bias.
2. Line 91: “Patients who fulfilled the American College of Rheumatology (ACR) criteria for RA” – the conformity with the ACR criteria is a pre-study determination or was it checked upon inclusion? Please include in the manuscript a short statement in either case (for example, you can report in the Results section how many patients did not fulfill the ACR criteria);
3. Line 91: “Patients who fulfilled the American College of Rheumatology (ACR) criteria for RA” – this means that the ACR criteria were the first exclusion criteria; therefore, please include it in the list of exclusion criteria;
4. Line 91: “were consecutively invited” – one can understand that, as they came to the hospital for RA-related issues, they were asked if they want to participate in the study? Is this the way you achieved inclusion randomness? This information should be briefly stated in the text. Also, how many invited patients declined to participate? You can report this in the Results section and consequently add refusal to participate in the list of exclusion criteria.
5. Line 101 “Demographic and Disease Characteristics” – in order to check classification criteria and respectively to calculate DAS28, you needed to collet data about rheumatoid factor status, anticitrullinated protein antibodies and acute phase reactants (erythrocyte sedimentation rate and C-reactive protein). Did you collect this data? If so, add them to the “Demographic and Disease Characteristics” list, alongside with brief information about how were they determined.
6. Line 101: “Demographic and Disease Characteristics” – reading this paragraph, one understands that the patients underwent clinical interview, clinical examination and questionnaires. Were all of these done in the same visit for each patient? A brief statement about this should be present in the text.
7. Line 102: “disease characteristic” – first, please use the plural (“characteristics”); second, did you collect information about RA treatment (glucocorticoids, csDMARDs, bDMARDs, tsDMARDs)? Treatment impacts disease activity and psychological state. Please update the text.
8. Line 104: “Disease activity was estimated” – to estimate the disease activity, you need to know how many joints are tender and how many are swollen. Did the authors perform the joint counts? Are the authors rheumatologists? How many rheumatologists counted the joints? Or, you did not and you just recorded this information from each attending rheumatologist in the hospital? This information is important since it impacts variance and measurement error and a brief statement about this should be present in the text.
9. Line 126 “The questionnaire was culturally adapted and translated into Chinese.” Did you did this for this study or did other researchers in the past did this? Judging from the consequent quote about convergent validity, the adaptation was done in another study. This is not clear form the text so please rephrase it.
10. Table 1: “DAS28” – what version of DAS28 is reported? The one calculated with ESR or with CRP?

Validity of the findings

no comment

Additional comments

no comment

Reviewer 3 ·

Basic reporting

See my comments

Experimental design

See my comments

Validity of the findings

See my comments

Additional comments

Thanks for opportunity to review manuscript entitled ‘‘Network analysis of depression and anxiety symptoms in Chinese rheumatoid arthritis patients’’ for Peerj Journal. The author/authors examined depression and anxiety symptoms in Chinese rheumatoid arthritis patients using network analysis. The strength of the manuscript includes examining variables of interest in a country where such studies are scarce. Overall, although the article is generally well written and deserves to be published in this journal some necessary and minor revisions must be made before the publication of the article. Because my main philosophy of reviewing a manuscript as reviewer and sometimes an editor to improve the manuscript and not punishing the authors, I provided very specific and detailed peer review of the manuscript to increase its quality and citation potential. I hope authors of the manuscript may benefit from my review. Necessary revisions reported section by section with the page and line number and when possible with suggestions.
Necessary Revisions
1. Conclusion section of abstract completely repetition of Results and must be revised.
2. One of the important weaknesses of limitation section is that authors did not give any information about importance of study about in their cultural context. Specifically, authors need to answer following question convincingly ‘‘Why it is important to examine depression and anxiety symptoms in rheumatoid arthritis patients using network analysis in Chinese cultural context? Perhaps, with cultural and practical reasons.
3. Authors used QoL as a outcome variable but did not give almost any information about it in introduction section. Authors must give compressive information about it and its possible connections. As well as previous studies examining this connection.
4. Add dot after disorders; instead of ;.
5. Authors did not give any demographic information in participants section. Authors must give at worst gender distribution, age (mean, sd, minimum, maximum), disease duration (mean, sd, minimum, maximum).
6. Participants section contain information about procedure and research design. Authors must separate this information or rename it as Participants and Procedure.
7. Please provide internal consistency coefficients for scales calculated in your study The Hospital Anxiety and Depression Scale and Short Form 36 health survey (SF-36).
8. No need subtitles in staticscal analysis section remove Network analysis.
9. Remove this ‘ ‘ from this line 127 and 128 ‘ ‘ Convergent validity and discriminant validity were satisfactory for all except the social functioning scale. Cronbachís ³ coefficient ranged from 0.72 to 0.88 except 0.39 for the SF scale and 0.66 for the vitality scale’’
10. Practical implications are completely missing and must be added to discussion.
11. Statistical analyses are correct.
12. Line 256 in previous study must be previous studies.

---

## Round 0.2 · accepted · Accept

In my opinion, this manuscript has been revised with attention to the reviewers' comments and can now be published.

Reviewer 1 ·

Basic reporting

The authors have addressed my comments.

Experimental design

The authors have addressed my comments.

Validity of the findings

The authors have addressed my comments.

Additional comments

The authors have addressed my comments.

·

Basic reporting

The authors have appropriately addressed the raised issues.

Experimental design

The authors have appropriately addressed the raised issues.

Validity of the findings

The authors have appropriately addressed the raised issues.

Additional comments

The authors have appropriately addressed the raised issues.

Reviewer 3 ·

Basic reporting

See my comments.

Experimental design

See my comments.

Validity of the findings

See my comments.

Additional comments

Thanks for opportunity re-review manuscript entitled ''Network analysis of depression and anxiety symptoms in Chinese rheumatoid arthritis patients.'' I have carefully reviewed the revised manuscript, and I am pleased to see that authors have addressed my previous comments in a thoughtful and comprehensive manner.The revised manuscript is now well-written, clear, and concise. The significance and novelty of work are well-established, and the results are presented in a clear and logical manner. I particularly appreciate the major and minor changes. Overall, I have no further comments or suggestions.